# Allogeneic Umbilical Cord-Blood-Derived Mesenchymal Stem Cells and Hyaluronate Composite Combined with High Tibial Osteotomy for Medial Knee Osteoarthritis with Full-Thickness Cartilage Defects

**DOI:** 10.3390/medicina59010148

**Published:** 2023-01-11

**Authors:** Yong-Beom Park, Han-Jun Lee, Hyun-Cheul Nam, Jung-Gwan Park

**Affiliations:** 1Department of Orthopedic Surgery, Chung-Ang University Gwangmyeong Hospital, Chung-Ang University College of Medicine, Seoul 06911, Republic of Korea; 2Department of Orthopedic Surgery, Chung-Ang University Hospital, Chung-Ang University College of Medicine, Seoul 06973, Republic of Korea; 3Department of Orthopedic Surgery, Madisesang Hospital, Seoul 02038, Republic of Korea

**Keywords:** osteoarthritis, cartilage, high tibial osteotomy, stem cells, umbilical cord blood

## Abstract

*Background and Objectives*: Although the effects of cartilage repair in patients who are undergoing high tibial osteotomy (HTO) remains controversial, cartilage repair may be required for the full-thickness cartilage defect because of a concern of lower clinical outcome. The purpose of this study was to investigate clinical outcome and cartilage repair following implantation of allogeneic umbilical cord-blood-derived MSCs (UCB-MSCs)–hyaluronate composite in patients who received HTO for medial knee osteoarthritis (OA) with full-thickness cartilage defect. *Materials and Methods*: Inclusion criteria were patients with a medial knee OA, a full-thickness cartilage defect (International Cartilage Repair Society (ICRS) grade IV) ≥  3 cm^2^ of the medial femoral condyle, and a varus deformity  ≥  5°. The full-thickness cartilage defect was treated with implantation of an allogeneic UCB-MSCs–hyaluronate composite following medial open-wedge HTO. Visual analogue scale for pain and Western Ontario and McMaster Universities Osteoarthritis Index (WOMAC) score were assessed at each follow-up. Cartilage repair was assessed by the ICRS cartilage repair assessment system at second-look arthroscopy when the plate was removed. *Results*: Twelve patients (mean age 56.1 years; mean defect size: 4.5 cm^2^) were included, and 10 patients underwent second-look arthroscopy during plate removal after a minimum of 1 year after the HTO. At the final follow-up of mean 2.9 years (range; 1–6 years), all clinical outcomes had improved. At second-look arthroscopy, repaired tissue was observed in all cases. One case (10%) showed grade I, seven (70%) cases showed grade II, and two (20%) cases showed grade III according to ICRS cartilage repair assessment system, which meant that 80% showed an overall repair assessment of “normal” or “nearly normal”. *Conclusion*: Allogeneic UCB-MSCs-HA composite implantation combined with HTO resulted in favorable clinical outcome and cartilage repair in all cases. These findings suggest that UCB-MSCs-HA composite implantation combined with HTO would be a good therapeutic option for patients with knee OA and full-thickness cartilage defects.

## 1. Introduction

High tibial osteotomy (HTO) is a well-established treatment option in knee osteoarthritis (OA) with varus deformity [1,2,3]. Shifting the mechanical axis of the lower extremity to the lateral side decreases the contact pressure on the medial compartment, which can provide the biological environment to prevent further degenerative changes [4,5]. Although several studies reported favorable short-term and mid-term outcomes after HTO, favorable long-term outcomes may be associate with adequate cartilage repair [6,7]. 

Although several therapeutic approaches to restore the cartilage have been investigated, the currently available options are not optimal for cartilage repair in knee OA. For the treatment of osteoarthritic cartilage defects, however, microfracture has been used [8,9], but it showed the deterioration of clinical outcomes over time [10]. 

Recently, mesenchymal stem cells (MSCs) have been gaining attention as a potential cell source for cartilage repair in older patients because of their unique properties, including immunomodulatory and anti-inflammatory capacities, and paracrine activity [11,12,13,14,15,16]. Allogeneic umbilical cord-blood-derived MSCs (UCB-MSCs) have advantages of non-invasive cell collection, high expansion capacity, hypo-immunogenicity, and immunomodulatory capacity [17,18,19]. Allogeneic UCB-MSCs implantation showed the safety and efficacy of cartilage repair in older patients with knee OA [20,21,22,23,24]. However, clinical outcomes and cartilage repair after allogeneic UCB-MSCs implantation combined with HTO have been reported rarely [25,26,27].

The purpose of this study was to investigate clinical outcome and cartilage repair following implantation of allogeneic UCB-MSCs–hyaluronate (UCB-MSCs-HA) composite combined with HTO for older patients who had medial knee OA with large full-thickness cartilage defect. It was hypothesized that an implantation of UCB-MSCs-HA composite would result in good cartilage remodeling and favorable clinical outcomes. 

## 2. Materials and Methods

A total of 62 patients who underwent HTO between 2016 and 2021 were retrospectively reviewed for study inclusion. HTO was indicated for patients who had isolated medial compartmental OA and varus malalignment of the lower extremity (varus deformity ≥ 5°) without ligament instability. Patients who underwent HTO and concomitant implantation of a UCB-MSCs-HA composite (Cartistem^®^) for medial compartmental OA and a full-thickness cartilage defect (International Cartilage Repair Society (ICRS) grade IV) ≥ 3 cm^2^ of the medial femoral condyle were included. Patients with a history of previous knee surgery, other cartilage repair procedure such as microfracture, and follow-up loss were excluded. Finally, 12 patients were included for this study (Figure 1). This study was approved by the institutional review board of our hospital (IRB No. 2109-025-19385).

## 3. UCB-MSCs-HA Composite

The medicinal product (UCB-MSCs-HA composite, Cartistem^®^) of this study was approved by the Korea Food and Drug Administration in 2012. Allogeneic UCB-MSCs were taken from donor UCB stored at a cord blood bank and were produced according to good manufacturing practice guidelines by Medipost (Seoul, South Korea). This product comprises 1.5 mL of UCB-MSCs (7.5 × 10^6^) and 4% HA.

## 4. Surgical Technique 

A standard arthroscopy was performed to assess cartilage defects, and arthroscopic procedures including debridement of the cartilage flaps, meniscectomy, or meniscal repair were performed if necessary. Biplanar open-wedge HTO was then performed according to preoperative planning to achieve a valgus alignment of 3–5°. 

After HTO, the UCB-MSCs-HA composite was implanted following the previously reported technique [20,21]. A small arthrotomy was made to expose the cartilage defect on the femoral condyle. Cartilage defects was prepared for healthy underlying bone and peripheral margin. Multiple drill holes (4 mm × 4 mm (diameter × depth)) in the subchondral bone were made to place the UCB-MSCs-HA composite. In addition, multiple drill holes with small diameter of 1.4 mm were made between the larger drill holes for better integration. The UCB-MSCs-HA composite was implanted into the drill holes carefully (Figure 2). 

## 5. Postoperative Rehabilitation

Venous impulse pumps were prescribed to prevent deep vein thrombosis. Quadriceps-strengthening and straight-leg-raising exercises were performed immediately after surgery. Additionally, a range-of-motion (ROM) exercise was allowed from postoperative day 1 and progressed as tolerated. Partial-weight bearing with crutch ambulation was allowed during 6 weeks, and full-weight bearing was allowed after 12 weeks. The second arthroscopy was performed during plate removal after the union of the osteotomy site.

## 6. Outcome Measures

Clinical and radiological evaluation were performed preoperatively at 1, 3, and 6 months; postoperatively at 1 year; and annually thereafter. A 100 mm visual analogue scale (VAS) for pain and Western Ontario and McMaster Universities Osteoarthritis Index (WOMAC) [28] were evaluated. Anteroposterior, lateral, and merchant views and whole-lower-extremity radiographs were obtained for radiological evaluation, including lower-limb alignment and Kellgren–Lawrence grade. The cartilage repair of the medial femoral condyle was assessed visually during second-look arthroscopy using ICRS Macroscopic Assessment of Cartilage Repair [29]. ICRS Macroscopic Assessment of Cartilage Repair consists of three items: degree of defect repair, integration to border zone, and macroscopic appearance, which is graded as the following: normal as grade I, nearly normal as grade II, abnormal as grade III, and severely abnormal as grade IV.

## 7. Statistical Analysis

VAS and WOMAC score for pain and Wilcoxon signed-rank test were used. All statistical analyses were executed using IBM SPSS statistics version 23.0 (IBM Corp., Armonk, NY, USA); a *p*-value < 0.05 was considered significant. 

## 8. Results

Twelve patients (9 women and 3 men) were included in this study. A mean age was 54.3 ± 7.8 years (range, 42–66 years). Seven patients had meniscal problems, which was treated by meniscectomy. The mean follow-up was 2.9 years (range, 1 to 6 years). The mean cartilage defect size of the medial femoral condyle was 4.5 cm^2^ (range, 4 to 6.9 cm^2^) (Table 1). 

At final follow-up, the VAS pain score was significantly improved from 61.6 to 11.4 and the WOMAC score from 46.6 to 12.3 (*p* < 0.05, Figure 3). The lower-limb alignment was changed from varus 6.7 to valgus 2.2.

At an average of 18 months after a UCB-MSCs-HA composite implantation combined with HTO, 10 patients underwent second-look arthroscopy during plate removal. At second-look arthroscopy, repaired tissue was observed in all cases. The mean total score of ICRS Macroscopic Assessment of Cartilage Repair was 8.4 (range, 5 to 12 points) (Table 2). One case (10%) showed grade I, seven (70%) cases showed grade II, and two (20%) cases showed grade III (Figure 4), which meant that 80% of the repaired cartilage was classified as “normal” or “nearly normal”.

## 9. Discussion

This study demonstrated that implantation of an allogeneic UCB-MSCs-HA composite combined with HTO provides favorable clinical outcome and cartilage restoration for full-thickness cartilage defects in knee OA with varus deformity. The consistent regenerative response was observed in all cases despite large full-thickness cartilage defects in knee OA, which may suggest that UCB-MSCs-HA implantation is required for better outcomes in patients who planned to undergo HTO for medial compartmental OA with large full-thickness cartilage defects and varus deformity. 

The repaired cartilage was observed in all cases during second-look arthroscopy after UCB-MSCs-HA composite implantation with concomitant HTO. A limited number of studies has reported that only HTO without cartilage repair procedure could provide cartilage repair potentially in some cases due to the change of the biomechanical environment [4,5]. However, the quality and quantity of the repaired cartilage were still insufficient. In addition, a randomized controlled trial regarding MSCs implantation combined with HTO reported a higher proportion of cartilage repair compared to HTO only [30]. A recent meta-analysis including four comparative studies reported that intra-articular MSCs administration combined with HTO showed better cartilage repair compared with the HTO alone [31]. In line with previous a meta-analysis, intra-articular UCB-MSCs-HA composite implantation combined with HTO in this study showed cartilage repair in all patients with medial compartmental OA with large full-thickness cartilage defects and varus deformity. Taken together with the results of this study and those of the meta-analysis, intra-articular MSCs administration could enhance cartilage repair in patients who underwent HTO for the treatment of knee OA and varus deformity.

Interestingly, regardless of the status of cartilage repair, pain and function at final follow-up was significantly improved compared to preoperative evaluation. Several studies have demonstrated that HTO could provide satisfactory pain and functional improvements [5,32,33], which may be induced by the decrease of contact pressure on the medial compartment by shifting the load from the medial to lateral compartment [4,5]. A recent meta-analysis study reported that intra-articular MSCs administration combined with HTO may improve clinical outcomes as compared to HTO alone [31]. In line with a previous meta-analysis, intra-articular UCB-MSCs-HA composite implantation combined with HTO in this study showed significant improvement in pain and function in all patients with knee OA with large full-thickness cartilage defects and varus deformity. Taken together, cartilage repair with MSCs administration in patients who underwent HTO could be a viable option for improved clinical outcomes at long-term follow-up. 

To date, there has been no reliable cartilage repair procedure for favorable outcomes in osteoarthritic cartilage defects [34]. Microfracture, the most common of a small cartilage defects, generally leads to fibrous repair tissue with unsatisfactory durability [8,9]. Autologous chondrocyte implantation (ACI) is usually recommended for younger patients with large focal chondral defects [35]. Both procedures are generally limited to restoring cartilage in large defects of older patients, with outcomes tending to deteriorate over time [10]. Some recent studies have demonstrated that surgical implantation of UCB-MSCs-HA composite could result in reliable cartilage repair in osteoarthritic cartilage defects. In this regard, surgical implantation of the UCB-MSCs-HA composite was selected for the cartilage repair procedure in this study. In accordance with previous studies with UCB-MSCs-HA composite, cartilage repair was observed in all cases despite full-thickness large cartilage defects more than 4 cm^2^. Therefore, surgical implantation of UCB-MSCs-HA composite could be a reliable option for cartilage repair in osteoarthritic knees. 

Some limitations of this study need to be addressed. First, this study was retrospective and did not include a control group. However, clinical outcome including cartilage repair after allogeneic UCB-MSCs implantation combined with HTO has rarely been reported [25,26,27]. Second, a small number of patients were included in this study. However, the regenerated cartilage was evaluated via second-look arthroscopy. In addition, only patients with a large full-thickness osteoarthritic cartilage defects were included in this study. Finally, magnetic resonance or biopsy for histological evaluation was not performed in this study, which would be a more reliable means for determining the properties of repaired cartilage. However, a direct visual evaluation of the repaired cartilage is one of the most reliable validated assessment tools for cartilage repair. 

In conclusion, this study showed that allogeneic UCB-MSCs-HA composite implantation combined with HTO resulted in favorable clinical outcome and cartilage repair in all cases. These findings suggest that UCB-MSCs-HA composite implantation combined with HTO would be a good therapeutic option for patients with knee OA and full-thickness large cartilage defects. 

## Figures and Tables

**Figure 1 medicina-59-00148-f001:**
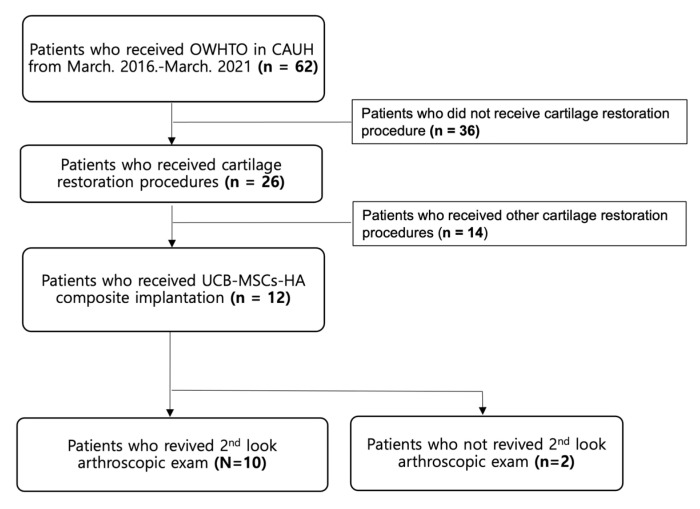
Flow diagram of the included patients.

**Figure 2 medicina-59-00148-f002:**
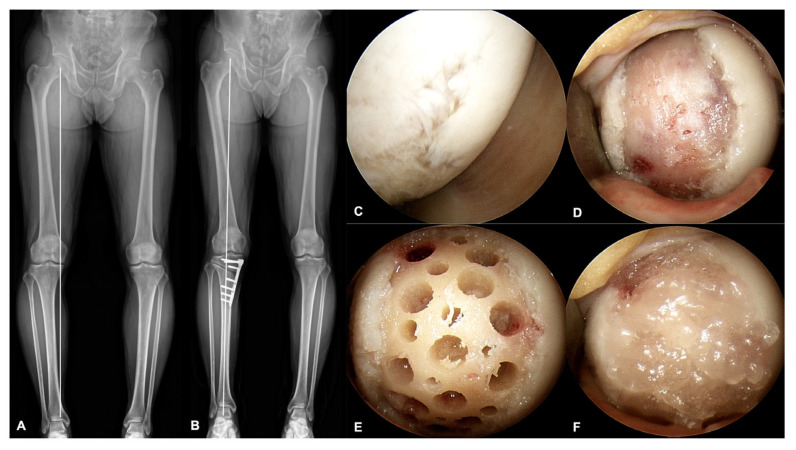
Surgical procedure of implantation for the allogeneic UCB-MSCs-HA composite and combined HTO. (**A**) Preoperative whole-lower-extremity radiograph showed varus limb alignment. (**B**) Postoperative whole-lower-extremity radiograph showed valgus limb alignment after HTO. (**C**) Arthroscopic inspection and confirmation of ICRS 3 and 4 cartilage defect. (**D**) Cartilage defect preparation for underlying healthy subchondral bone and peripheral margin. (**E**) Multiple drill holes in the cartilage defect site. (**F**) Implantation of the UCB-MSCs-HA composite into drill holes and cartilage defect surface (Cartistem^®^).

**Figure 3 medicina-59-00148-f003:**
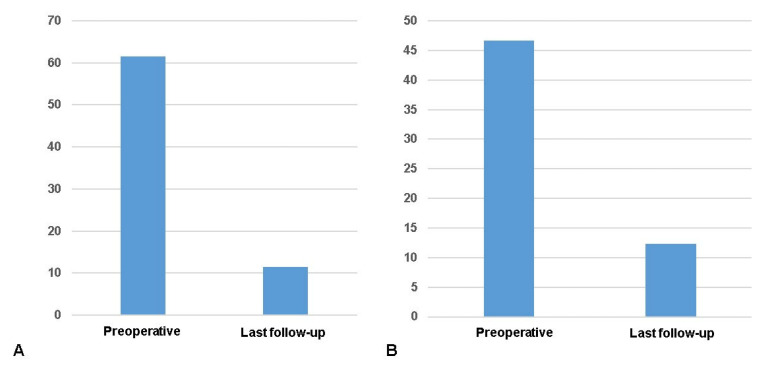
Clinical outcomes after the UCB-MSCs-HA composite implantation combined with HTO. (**A**) Pain on the 100 mm visual analogue scale (VAS) score; (**B**) Western Ontario and McMaster Universities Osteoarthritis Index (WOMAC) score.

**Figure 4 medicina-59-00148-f004:**
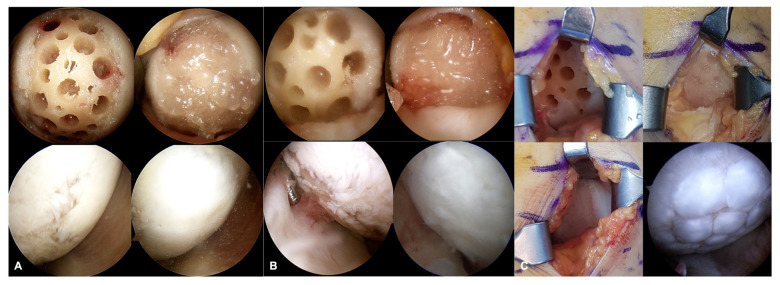
Evaluation of the repaired cartilage according to the ICRS Macroscopic Assessment of Cartilage Repair during second look arthroscopy. (**A**) A case of ICRS cartilage repair assessment score of 12, which was grade I in overall repair assessment, i.e., “normal”. (**B**) A case of ICRS cartilage repair assessment score of 9, which was grade II in overall repair assessment, i.e., “nearly normal”. (**C**) A case of ICRS cartilage repair assessment score of 5, which was grade II in overall repair assessment, i.e., “abnormal”.

**Table 1 medicina-59-00148-t001:** Demographics and Baseline Characteristics of patients of this study.

	UCB-MSCs-HA Composite
**Age** years (mean ± SD)	54.3 ± 7.8
**Sex***n* (%)	
Male	3 (25.0)
Female	9 (75.0)
**BMI** (body mass index) kg/m^2^ (mean ± SD)	25.9 ± 2.8
**HKA angle**	173.3 ± 2.8
**Osteoarthritis***n* (%)	
K-L grade I	0
K-L grade II	2 (16.7)
K-L grade III	9 (75.0)
K-L grade IV	1 (8.3)
**Pain on 100 mm VAS *** (mean ± SD)	61.6 ± 7.9
**WOMAC score ^†^** (mean ± SD)	46.6 ± 5.3
**Cartilage Defect Characteristics**	
**Size** cm^2^ (mean ± SD)	4.5 ± 1.0
**Location** *n* (%)	
**MFC**	9 (75.0)
**MFC and MTP**	3 (25.0)

Kellgren–Lawrence (K-L) grade II or III, sustaining typical bipolar lesions with varying degrees of severity. * Pain on the 100 mm visual analogue scale (VAS) ranges from 0 to 100, with higher score indicating worse results. † Western Ontario and McMaster Universities Osteoarthritis Index (WOMAC) ranges from 0 to 92, with higher score indicating worse results.

**Table 2 medicina-59-00148-t002:** Cartilage repair assessment of second arthroscopic findings.

	Score	Mean ± SD
**Degree of defect repair**		3.5 ± 0.5
Level with surrounding cartilage	4	
75% repair of defect depth	3	
50% repair of defect depth	2	
25% repair of defect depth	1	
0% repair of defect depth	0	
**Integration to the border zone**		2.5 ± 1.1
Complete integration with surrounding cartilage	4	
Demarcating border < 1 mm	3	
2/4 of graft integrated, 1/4 with a notable border >1 mm width	2	
1/2 of graft integrated with surrounding cartilage, 1/2 with a notable border >1 mm	1	
From no contact to 1/4 of graft integrated with surrounding cartilage	0	
**Macroscopic appearance**		2.4 ± 1.0
Intact smooth surface	4	
Fibrillated surface	3	
Small, scattered fissures or cracks	2	
Several small or few but large fissures	1	
Total degeneration of the grafted area	0	
**Total score** (mean ± SD)	0–12	8.4 ± 2.3
**Grading system ***		*n* (%)
Grade 1: Normal	12	1 (10)
Grade 2: Nearly normal	8–11	7 (70)
Grade 3: Abnormal	5–7	2 (20)
Grade 4: Severely abnormal	0–4	0 (0)

* Grade was classified according to the total score.

## Data Availability

Not applicable.

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
