# Peer review of "Allogeneic Umbilical Cord-Blood-Derived Mesenchymal Stem Cells and Hyaluronate Composite Combined with High Tibial Osteotomy for Medial Knee Osteoarthritis with Full-Thickness Cartilage Defects"

_medicina, 2023, doi:10.3390/medicina59010148_

Round 1
Reviewer 1 Report
thanks for submitting this paper
this is a small case series aimed to report the results of HTO+UCB-MSCs in patients with cartilage defects and limb malalignment.
overall well-written paper with some major methodological issues
main flaws of the study are (1) small sample size, (2) no control group, (3) short follow-up [1 year? please report standard deviation for FU time]
COMMENTS
-please be sure that the manuscript is revised by a proficient English speaker researcher
-please be sure that references are correctly formatted, relevant, and updated
-abstract: well written
-line 67: please report the study design
-figure 1 (you had 14 patients with cartilage procedures, why you chose not to include a control group?)
line 121 (how were radiological measurements performed and how reliable were these ?)
line 129 (please report sample size calculation)
results: well written
Reviewer 2 Report
This paper is well written . I have only one addition if they have 65 patients with HTO s why not compare the outcomes of the 12 with the 50 or so patients who did not have this procedure . it would make a great paper.
Author Response
Thank you for your comment. We know that the comparison with the control group makes this manuscript better. However, we focused on cartilage repair following novel MSCs product implantation in this study.
Someday, we will prepare the comparison study according to your comment.
Thank you.
Round 2
Reviewer 1 Report
thanks for your revision